# Community-like genome in single cells of the sulfur bacterium *Achromatium oxaliferum*

Danny Ionescu[1], Mina Bizic-Ionescu[1], Nicola De Maio[2], Heribert Cypionka[3] & Hans-Peter Grossart [1,4]

Polyploid bacteria are common, but the genetic and functional diversity resulting from polyploidy is unknown. Here we use single-cell genomics, metagenomics, single-cell amplicon sequencing, and fluorescence in situ hybridization, to show that individual cells of *Achromatium oxaliferum*, the world's biggest known freshwater bacterium, harbor genetic diversity typical of whole bacterial communities. The cells contain tens of transposable elements, which likely cause the unprecedented diversity that we observe in the sequence and synteny of genes. Given the high within-cell diversity of the usually conserved 16S ribosomal RNA gene, we suggest that gene conversion occurs in multiple, separated genomic hotspots. The ribosomal RNA distribution inside the cells hints to spatially differential gene expression. We also suggest that intracellular gene transfer may lead to extensive gene reshuffling and increased diversity.

[1] Leibniz Institute of Freshwater Ecology and Inland Fisheries, Department of Experimental Limnology, Alte Fischerhuette 2, 16775 Stechlin, Germany. [2] Institute for Emerging Infections, Oxford Martin School, University of Oxford, 34 Broad Street, Oxford OX1 3BD, UK. [3] Institute for Chemistry and Biology of the Marine Environment, University of Oldenburg, 26111 Oldenburg, Germany. [4] Institute of Biochemistry and Biology, Potsdam University, 14476 Potsdam, Germany. Correspondence and requests for materials should be addressed to D.I. (email: ionescu@igb-berlin.de) or to H.-P.G. (email: hgrossart@igb-berlin.de)

Polyploidy, the condition of having multiple chromosome copies per cell is a frequent phenomenon in eukaryotic organisms[1]. Polyploidy is suggested[2, 3] and in some cases shown[4] to be advantageous in regulation of gene expression, DNA repair, and supporting large cell sizes. Despite most commonly studied bacteria being haploid[1], polyploid *Archaea* and *Bacteria* (defined as having more than 10 genome copies) are common and can contain up to thousands[5] of genome copies (hereafter chromosomes). These chromosomes are believed to be nearly identical copies and safeguarded against mutations by gene conversion (asymmetrical homologous recombination resulting in one allele "overwriting" another)[6].

Polyploidy has been suggested to have a major role in the evolution of eukaryotes by allowing genomic rearrangements and gene duplication[7, 8] that eventually result in different functionality by similar organisms. In *Bacteria* and *Archaea* the significance of polyploidy has received less attention. Polyploidy can lead to divergence of the coding material allowing the cells to experiment with new gene/protein versions[5, 9]. Thus, a polyploid bacterium with divergent genome copies would benefit from the genetic diversity of a colony within each single cell[9]. However, observations from the highly polyploid *Epulopiscium* spp. using marker genes and recently from genomic studies of *Candidatus* Marithrix sp., suggested that genomic copies within a cell are all extremely similar[9, 10], possibly as a consequence of strong gene conversion, within-cell genome population bottlenecks at reproduction, and limited between cell recombination.

*Achromatium* sp. is the largest known unicellular freshwater bacterium, with several described size classes reaching up to 15 × 125 μm[11, 12]. It is a colorless sulfur-oxidizing bacterium typically found at the oxic–anoxic interface in sediments of temperate freshwater lakes[11]. The cells contain large calcite bodies and sulfur granules[12, 13]. *Achromatium* was mostly studied in freshwater environments with several species and phylotypes

described[14], but may be found in tidal salt marsh[12] and in mineral springs[15] as well. According to nucleic acid staining[12, 16], like other large sulfur bacteria[17], *Achromatium* appears to be polyploid.

Here we study *Achromatium* cells using genomic and metagenomic data from single and pooled "hand-picked" *Achromatium oxaliferum* cells from Lake Stechlin, NE Germany, coupled with 16S ribosomal RNA (rRNA) analysis of 27 single *Achromatium* cells and fluorescence in situ hybridization (FISH). We find extreme intracellular genetic diversity, and suggest that *Achromatium* undergoes intracellular gene duplications, re-assortments, and divergence with reduced or minimal gene convergence, leading to genetic diversity typical for populations rather than single cells. Our data suggests that the cells are equipped with numerous transposases, insertion sequences, and DNA editing factors as the machinery responsible for the intracellular evolution. These processes could explain the highly geneticallyheterogeneous *Achromatium* population at the level of individual cells.

## Results

**Evidence of polyploidy.** A light micrograph of a dividing *Achromatium* cell from Lake Stechlin overlaid with the parallel DNA staining image (Fig. 1, Supplementary Fig. 1) shows that the individual cells contain multiple DNA spots that are not localized in one single area but rather spread across the cell, mostly in between calcium carbonate bodies. Analysis of several cells showed an average of 199 ± 46 spots. Given that the spots had varying fluorescence intensity, we cannot rule out each spot containing a varying amount of DNA[18], i.e., a different number of chromosomes or chromosomes of varying sizes. Based on previous knowledge on large sulfur bacteria and giant *Firmicutes*[17] these multiple DNA spots confirm the polyploid nature of *Achromatium*.

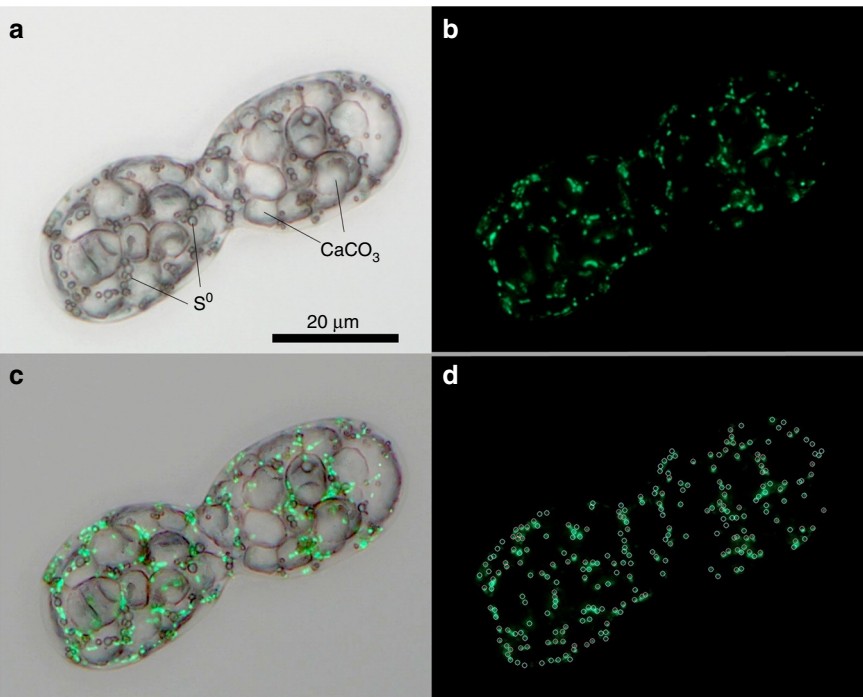

**Fig. 1** Dividing cell of *Achromatium* sp. **a** Bright field showing calcite crystals (CaCO₃) and sulfur droplets (S⁰). **b** Nucleic acids stained by SybrGreen I in the same cell. **c** Overlay of **a** and **b** showing that sulfur and nucleic acids spots are present in the grooves around the calcites, but not at the same positions. **d** Count of 244 DNA spots using the software tool CountThem. Both bright field and fluorescence images were taken as focus stacks of 22 images covering the full-cell depth and were processed by the stacking program PICOLAY. A similar image stained with the DNA exclusive dye picoGreen, is provided as Supplementary Fig. 1

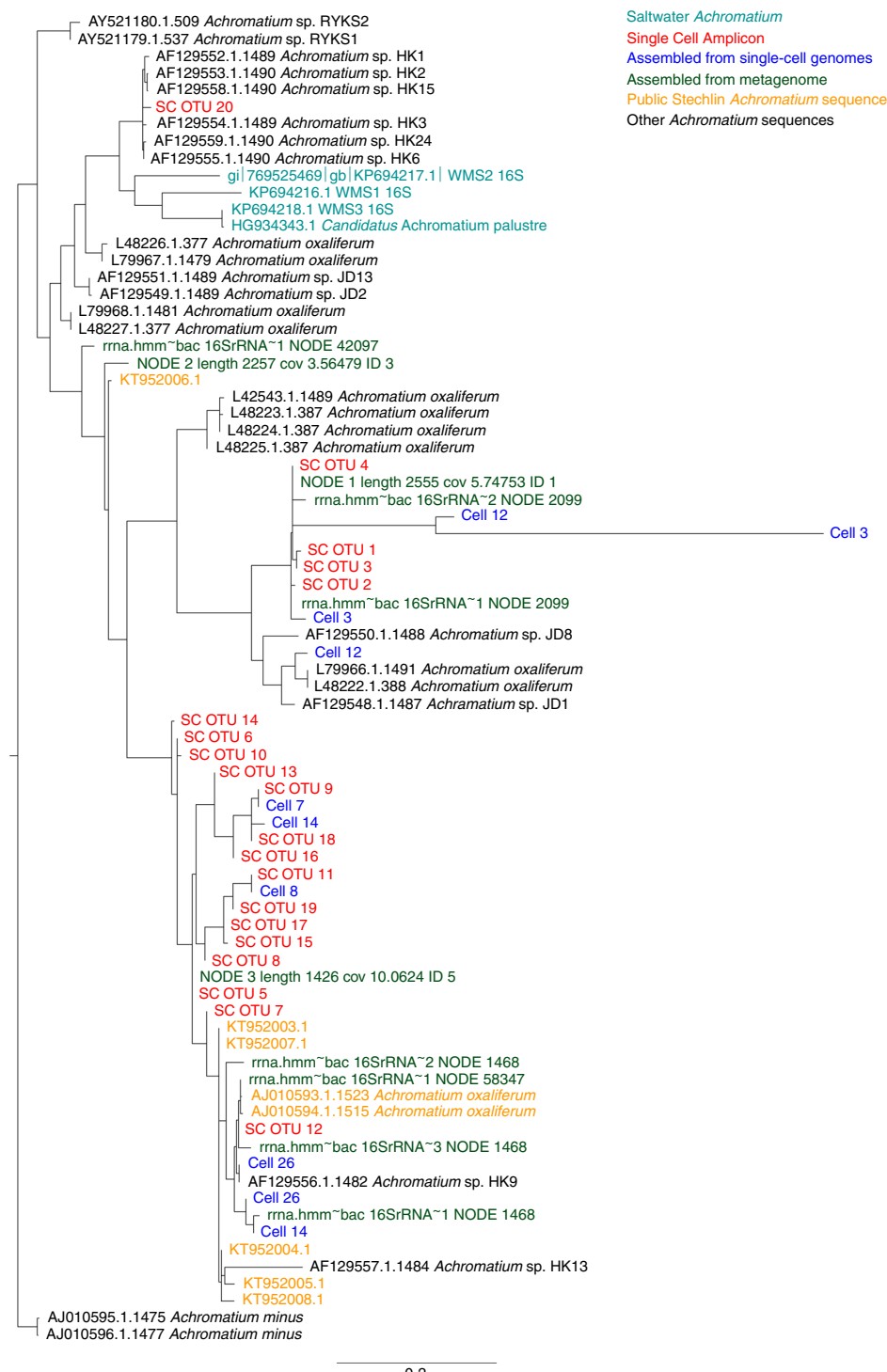

**Fig. 2** Maximum likelihood tree of 16S rRNA sequences from *Achromatium*. The sequences were obtained from single *Achromatium* cells from Lake Stechlin, metagenomics data of the same cell population, and reference sequences. A similar tree which includes distance-clustered amplicon sequences is given in Supplementary Fig. 2A. A similar tree created using only full-length sequences to which the shorter ones were added by parsimony is provided as Supplementary Fig. 3

**Community-like rRNA diversity in single cells of *Achromatium*.** Metagenomic data obtained from sequencing of ca. 10,000 "hand-picked" and well pre-washed *Achromatium oxaliferum* cells from Lake Stechlin were analyzed for the presence of 16S rRNA gene sequences. Most of the 16S rRNA gene reads (>98%) were associated to *Achromatium* sp., suggesting a low level of contamination by the remaining epibiotic bacteria. These reads assembled into three different full-length 16S rRNA sequences (93–95% similarity; Fig. 2, Supplementary Figs. 2 and 3). Several additional *Achromatium* affiliated partial reads were also identified (>91% identity) in the metagenome assembled data (Fig. 2, Supplementary Figs. 2 and 3).

A section of the 16S rRNA gene was further sequenced from 27 individual *Achromatium* cells. The V1–V4 region was sequenced

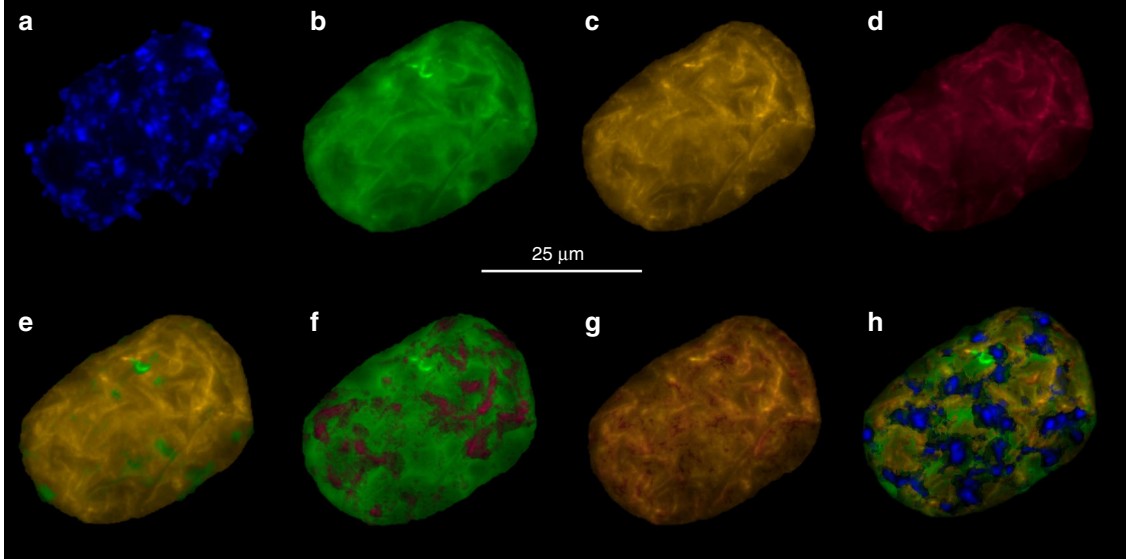

**Fig. 3** Fluorescence in situ hybridization. Fluorescence in situ hybridization of a DAPI stained **a** *Achromatium* cell using probes AchroCLII-IV (5′-cgatcgtcgccttggtaggctt-3′) **b**, AchroCLIII (5′-cgatcgttgccttggtgggctt-3′) **c**, and AchroCLI (5′-ggatcgtcgccttggtaggcca-3′) **d**. The used three different probes for *Achromatium* match the clusters in Supplementary Fig. 2A. **e–g** Show overlaid images of 2 probes combinations. An overlay of all probes as well as the DAPI staining is shown in **h**. Results from additional cells are shown in Supplementary Fig. 6

for 5 of these cells, while for the reminder the V5 region was sequenced as a test of contamination prior to single-cell genome sequencing. Both sets were analyzed using the stringent DADA2 R package[19] resulting in 20 and 177 sequence variants, respectively (Fig. 2 and Supplementary Figs. 2–4). Distance based clustering of the same sequences at an identity cutoff of 97% resulted in 1189 and 6909 operational taxonomic units (OTUs), respectively.

We sequenced the genomes of six cells, which resulted in more than 99% *Achromatium* related 16S rRNA gene sequences. From these cells, 10 full or partial 16S rRNA gene sequences were obtained (Fig. 2, Supplementary Fig. 3).

A maximum likelihood phylogenetic analysis of all OTUs overlapping the V1–V4 region alongside previously published *Achromatium* sequences shows no cell-specific clusters (Fig. 2, Supplementary Figs. 2–4). Two earlier sequences reported from Lake Stechlin and assigned to two different species[20] appear both as members of a single cluster (cluster 4; Supplementary Fig. 2). This suggests that these sequences belong to members of a single group with elevated genetic diversity. The alignment of the rRNA gene sequences shows that the genetic diversity is concentrated in the hypervariable regions, supporting the idea that this is not the result of random sequencing errors, but of an evolutionary process (Supplementary Fig. 5). This is surprising as the rRNA genes are generally believed to undergo gene conversion resulting in concerted evolution within species[21, 22].

**Spatially differential expression of rRNA.** To confirm the presence and expression of several different rRNA gene alleles in individual cells, two sets of FISH probes were designed: the first based on complete 16S rRNA sequences as assembled from the metagenome, and the second based on the clustering of the V1–V4 amplicons obtained from single cells (Supplementary Fig. 2A). Different cells showed positive signals for 1, 2, or 3 probes of either sets (Fig. 3, Supplementary Figs. 2B and 6), confirming that different *Achromatium* rRNA sequences do not originate from different species but from intracellular diversity. Previous FISH analyses of *Achromatium* from Lake Stechlin have similarly identified cells labeled by one or more probes

designed to target what were thought to be different species[16]. It is not unprecedented that bacteria contain multiple and different copies of the rRNA operon with dissimilarities between 16S rRNA genes in one cell up to 11%[23]. The FISH signal of the different rRNAs overlapped only partially (Fig. 3, Supplementary Figs. 2B and 6) suggesting that at least in some cases different alleles of the same genes may be expressed in different locations in the cell.

**Achromatium cells harbor multiple non-identical chromosomes.** The assembly of single-cell genomes (6 × ~ 12,000,000 reads) and metagenomic data (~ 96,000,000 reads) resulted in relatively short contigs (< 82,000 nt and < 56,000 nt long, respectively). Given the almost absolute presence of *Achromatium* sequences among the 16S rRNA genomic (single cell) and metagenomic reads, the elevated read output would have been sufficient to robustly cover even a large prokaryotic genome (e.g., ~2000-fold coverage of a single 7 Mbp genome). The estimated size of the *Achromatium* genome based on the 6 single cells and the various metagenomic bins ranges between 3.5 and 12 Mbp (Supplementary Data 1). Therefore, we suggest that *Achromatium* does not harbor many identical copies of a single genome (like, e.g., *Epilopiscium*) but rather many diverse chromosomes. This is also supported by the multiple different copies of many of the genes (see below). We hypothesize that our assembled contigs represent the relatively conserved regions in an otherwise highly diverse inter- and intra-cellular (polyploid) chromosomal environment. The genome regions connecting the assembled genes probably vary significantly between chromosomes, resulting in a low sequencing coverage per-variant, insufficient for assembly.

Tetranucleotide binning[24, 25] coupled to GC distribution and assembly coverage are often used in extracting individual genomes out of metagenomic data and are referred to as binning. The completion and homogeneity of each bin (genome) is evaluated by the percent presence of general or lineage specific sets of "single-copy" marker genes. We have used this technique to evaluate the presence of multiple *Achromatium* species in our samples, each containing a unique genome. Binning of

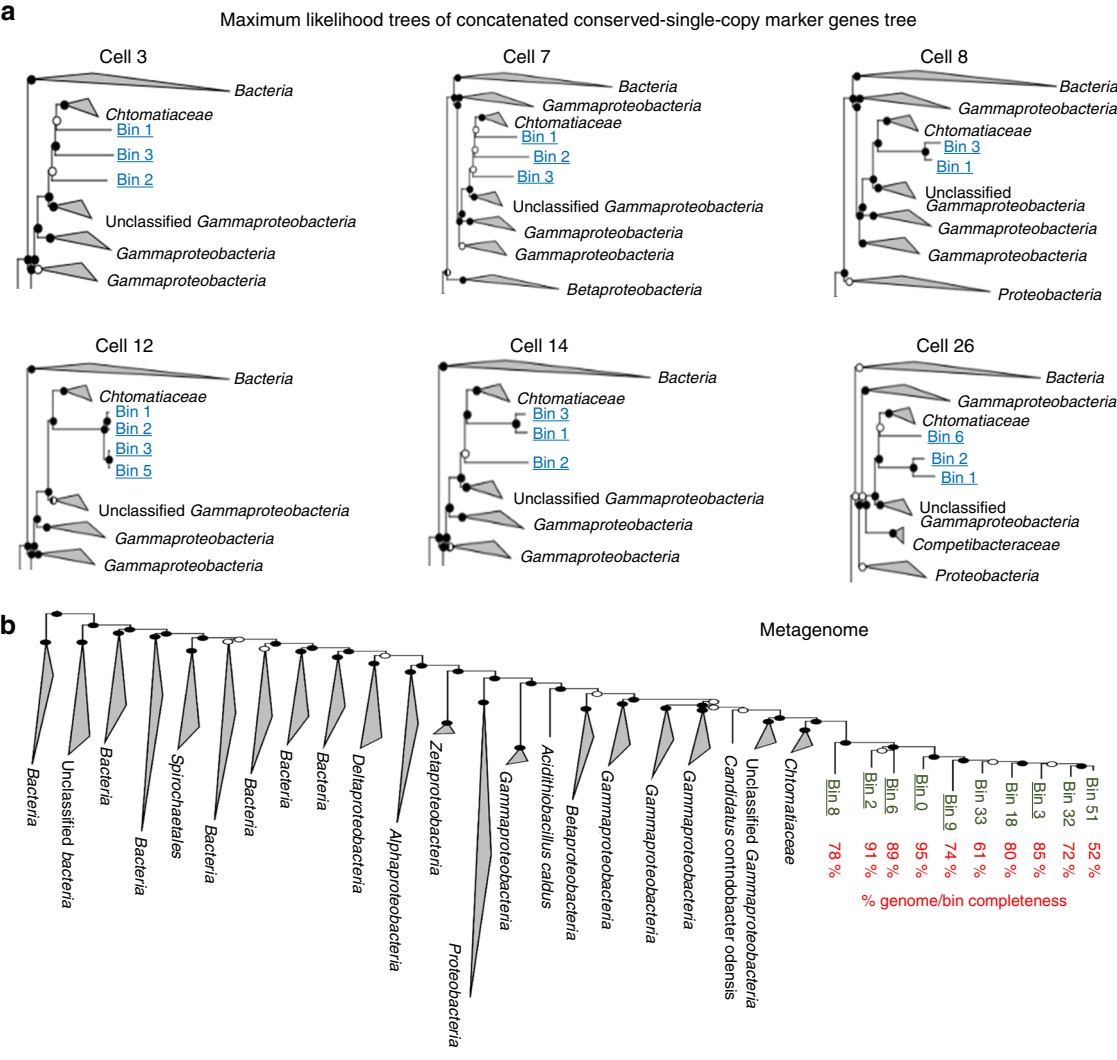

**Fig. 4** Maximum likelihood trees calculated from concatenated sequences of conserved single copy marker genes. The genes were found in genes of the 6 sequenced single cells **a** and the different genomic bins of the *Achromatium* metagenome **b**

**Table 1 Abundance of various genetic elements in *Achromatium* genomes and metagenomes**

| | Freshwater/Stechlin | | | | | | | Saltwater | | | |
| --- | --- | --- | --- | --- | --- | --- | --- | --- | --- | --- | --- |
| | Cell 3 | Cell 7 | Cell 8 | Cell 12 | Cell 14 | Cell 26 | MGM | *A. palustre* | WMS1 | WMS2 | WMS3 |
| Insertion sequences | 160 | 145 | 49 | 36 | 383 | 88 | 545 | 27 | 20 | 15 | 85 |
| Transposases (Prokka annotation) | 51 | 47 | 50 | 51 | 93 | 29 | 546 | 19 | 3 | 12 | 27 |
| Origin of replication | 1 | 6 | 3 | 1 | 3 | 4 | 19 | 5 | 3 | 5 | 5 |

either single-cell genomes or metagenomic data results in multiple bins with a varying degree of completion (<42% for single cells and 52–95% for the metagenome; Supplementary Data 1). Both single cells and metagenomic bins contain multiple copies of some "single copy" marker genes (Supplementary Data 2) with predicted (gene set) duplication levels up to 1.5 times (of 138 genes analyzed; Supplementary Data 1). A similar approach applied to the genomes of four published single *Achromatium* cells from brackish/marine environments (WMS1-3[15] and "*Candidatus* Achromatium palustre"[12] divided the assembled data into 2–4 bins without overlapping marker genes (Supplementary Data 3).

The presence of multiple bins as well as the outcome of phylogenetic trees of concatenated "single copy" marker genes[26] assigned to each bin obtained from the metagenomic or single-cell assemblies could suggest the possible presence of multiple *Achromatium* species whose different genomes can be separated by sequence patterns (Fig. 4). However, several facts rule out this possibility. First, we are confident that single-cell genomes contain only one *Achromatium* cell. Second, the multiple 16S rRNA gene alleles resolved in the metagenome were later found in the single-cell genomes, as well as in the specific amplicon sequences obtained from our single cells. Third, all single-cell genomes contain multiple and different copies of some of the "single copy" marker genes, further supporting our hypothesis of elevated intracellular diversity. This as well confirms that multiple copies of "single copy" marker genes in the metagenomic data must not represent multiple and different

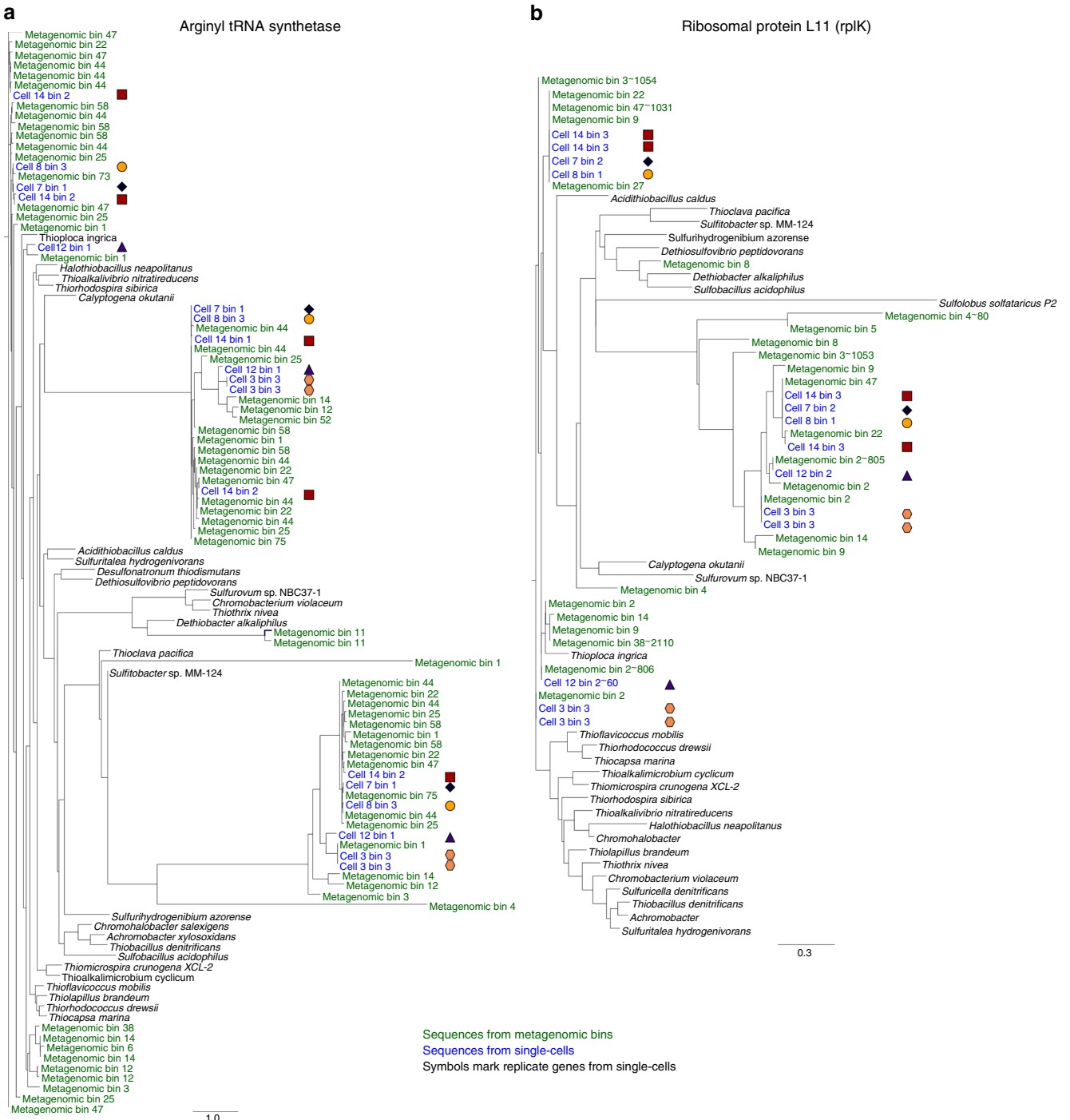

**Fig. 5** Maximum likelihood tree of two proteins believed to be single-copy marker genes. The genes encode Arg-tRNA-synthase **a** and ribosomal protein L11 (rplK) **b**. Replicate copies of the genes identified in the same cells are marked by identical shapes. Additional trees of "single copy" marker genes (protein trees) are given in Supplementary Data 4

genomes. Fourth, all metagenomic bins were associated to *Achromatium*, but comparing trees of individual marker genes from different bins showed phylogenetic inconsistencies (i.e., dissimilarity between the trees; Supplementary Data 4). Hence, we hypothesize that the separation into individual genomic/metagenomic bins is a by-product of sequence divergence across chromosomes, and not a consequence of the presence of multiple separate species. Mobile genetic elements such as transposons often use tetranucleotide recognition sites which are disrupted or duplicated upon insertion[27–29]. The Lake

Stechlin *Achromatium* genomes and metagenome are extremely rich (>90 and >500, respectively) in transposases (Table 1). These are likely responsible for generating diverse tetranucleotide patterns (Fig. 5).

Phylogenetic analysis of "single copy" marker genes shows that individual cells harbor multiple and different copies (e.g., arg-tRNA-synthetase and *rplK* in Fig. 4a, b; for more genes see Supplementary Data 5). The *Achromatium* metagenome further reveals that both inter- and intracellular diversity cover the diversity observed in the overall community.

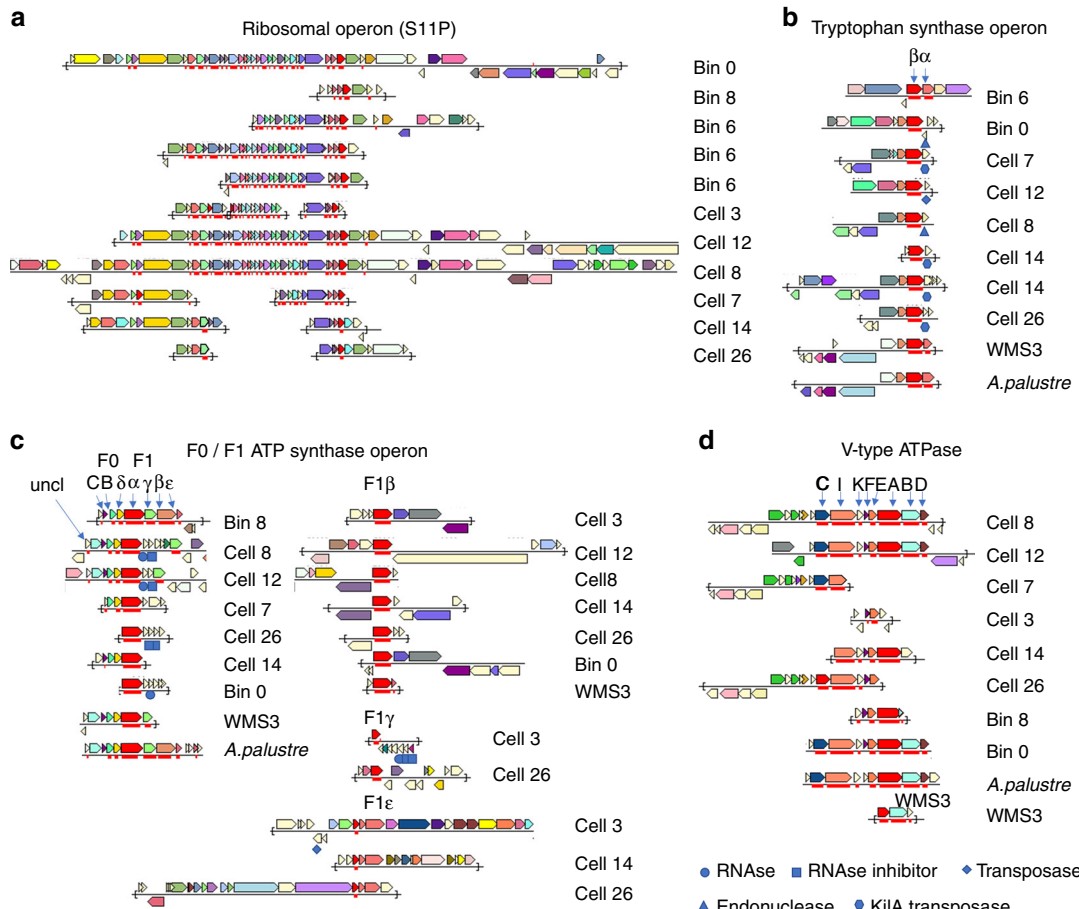

**Fig. 6** Gene synteny of four bacterial operons. The operons are: ribosomal **a**, tryptophan synthase **b**, ATP synthase **c**, and V-type ATPase **d**. Additional individual genes from the tryptophan synthase and ATP synthase operons were identified and are not shown here. The data are available on IMG genome IDs 2642422595-9, 2710724216-17 and 2711768587-90

**Gene synteny**. Gene synteny can be of crucial significance to functionality. While the relocation of individual genes within the genome or between multiple copies of the genome may not have damaging effects, the disruption of operons may be fatal. As such we looked into the gene synteny of the ribosomal protein operons, the tryptophan synthase, the ATP synthase, and the V-type ATPase (Fig. 6), as well as solitary genes such as *recA* (Supplementary Fig. 7).

The gene neighborhood of solitary genes such as *recA* differs between most copies (Supplementary Fig. 7). In contrast, the ribosomal operon gene neighborhood is highly conserved (Fig. 6a). The sequence similarity between copies of ribosomal proteins (e.g., *rplK* Fig. 4b) is higher than in non-operon-based marker genes (Fig. 3b, c, Supplementary Data 6), as previously shown[30]. This suggests that the conservation pressure differs between different proteins strengthening our hypothesis that the observed phenomenon of intracellular genomic diversity is real and not a methodological artifact. It appears though that not all known operons are fully conserved. The tryptophan synthase and ATP-synthase operons (Fig. 6b, c) have been recovered only in one case as a full set of genes (Fig. 6c). In most other cases the operon was interrupted by known transposable elements, by a gene set consisting of an RNAse and RNAse inhibitor, or by an endonuclease. Additional single genes belonging to these operons occur individually on other scaffolds; data not shown, available on IMG[31] with genome IDs 2642422595-9, 2710724216-17, and 2711768587-90. In contrast, the operon of the V-type ATPase seems to be strongly conserved. The effect this has on the

*Achromatium* functionality is currently not clear. Suzuki et al.[32] have shown that for the F0F1 ATP synthase, the separate yet simultaneous expression of some of the genes maintains functionality. Thus, while some operons may be interrupted, they may still function if maintained under similar regulatory factors.

**Large genetic diversity within and between *Achromatium* cells**. Based on the above results we consider the entire set of assembled sequences associated to *Achromatium* sp. as a "community genome". A pan-genome comprising all metagenomic sequences attributed to *Achromatium*-associated bins consisted of >3500 proteins, excluding hypothetical genes (Supplementary Data 7). We used two tests to investigate if different proteins are under different evolutionary pressure. In the first test, we calculated for each and across all single-cell genomes the average amino acid distance between copies of proteins that occur more than twice (79–296 proteins, median = 90; Fig. 7a and Supplementary Fig. 8) and for the metagenome the distance between those occurring more than five times ( ~ 1400 proteins, see Fig. 7b; Supplementary Data 7). In the second test, we calculated from gene copy alignments the average ratio between non-synonymous and synonymous mutations (Ka/Ks) for ~1180 of the above metagenomic proteins which could be annotated using Seed Subsystem[33] (Fig. 7c). Among the single cells, 25% of the proteins have a Dayhoff distance[34] smaller than one substitution per site (Fig. 7a). With metagenomic data this number increases to 50% (Fig. 7b) with no correlation between gene copy number and

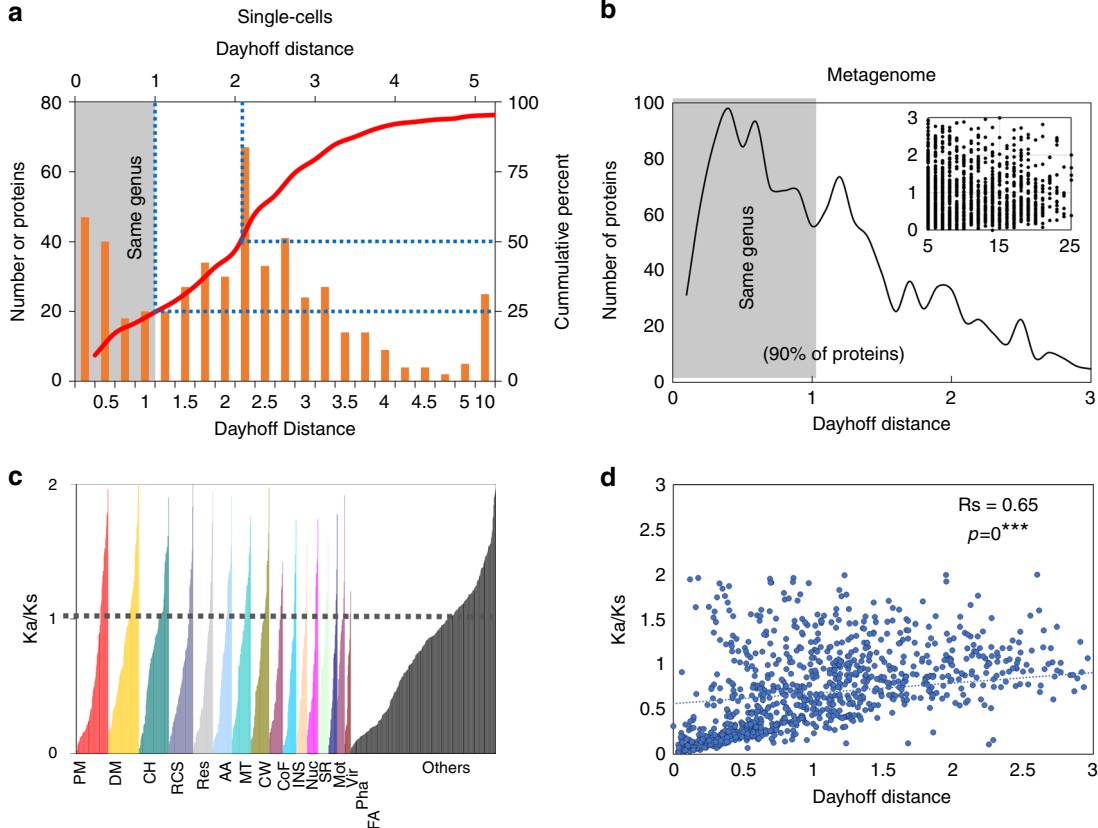

**Fig. 7** Distances between multiple copies of the same protein and conservation pressure. **a** Protein numbers ($n = 505$) distribution along the average Dayhoff distance across single cells, between proteins with multiple copies per single cells. The curve shows the cumulative percent of total proteins. A similar distribution across all single cells is shown in Supplementary Fig. 8. **b** Protein numbers ($n = 1400$) distribution along the Dayhoff distance (0–3), showing a decreasing number of proteins as the distance increases. The insert shows the lack of correlation between distance (*Y* axis) and gene copy number (*X* axis). **c** Ratio between non-synonymous and synonymous mutations (Ka/Ks) calculated for 1040 proteins grouped based on the first level of the seed subsystem[32] hierarchical system. The dashed line at a ratio of 1 marks proteins inferred under conservation pressure (Ka/Ks < 1), stable (Ka/Ks = 1), and evolving (Ka/Ks>1). The seed groups are labeled as follows: *AA* amino acids and derivatives; *CH* carbohydrates; *CoF* cofactors, vitamins, prosthetic groups, pigments; *CW* cell wall and capsule; *DM* DNA metabolism; *FA* fatty acids, lipids, and isoprenoids; *INS* iron, nitrogen and sulfur metabolism (small groups combined for plotting purposes); *Mot* motility and chemotaxis, *MT* membrane Transport; *Nuc* nucleosides and nucleotides; *Others* smaller groups and non-classifiable proteins; *Pha* phages, prophages, transposable elements, Plasmids; *PM* protein metabolism; *RCS* regulation and cell signaling; *Res* respiration; *SR* stress response; *Vir* virulence, disease and defense. **d** Correlation between protein distance and the Ka/Ks ratio (Spearman Rs = 0.64)

distance (Fig. 7b, insert). This highlights the high genetic diversity within the community which can be even higher when looking at its individual members. A distance of one substitution per amino acid site has been shown to be at the upper limit of protein distance between species of the same genus. However, a distance of three amino acids substitutions per site would normally suggest a lower phylogenetic relation[35]. Interestingly, there is not a big difference between overall diversity patterns among single cells and within individuals (Fig. 4a and Supplementary Fig. 8). The salt water *Achromatium* cells/populations have a similar broad diversity among the multiple-copy proteins (Supplementary Fig. 8), despite having single rRNA alleles rather than multiple ones as our freshwater cells have. The Ka/Ks ratio is lower than 0.5 for 410/1180 proteins (lower than 1.0 for 775/1180 proteins), i.e., under evolutionary pressure for conservation. No correlation was found between the Ka/Ks ratio and gene copy number. Hierarchical classification of the proteins based on the Seed Subsystem[33] classification (Level 1) shows no group specific trends. Overall, no specific protein group seems under stronger or weaker conservation pressure. The correlation between the Ka/Ks ratio and the averaged protein distance (Fig. 7d) confirms that this is not a sequencing artifact.

Using freshwater and marine single-cell genomes as well as metagenomic bins (Fig. 4a) we conducted amino acid identity and average nucleotide identity analyses[36]. Overlaying the results (Supplementary Fig. 9) with the thresholds of Rodriguez-R and Konstantinidis[37] places most of freshwater genomes and bins as different species in the same genus (similarly as when compared to the salt water cells). Nevertheless, all the data presented thus far suggest that the genetic diversity of freshwater *Achromatium* from Lake Stechlin does not originate from multiple species. Instead, we propose that each individual *Achromatium* cell harbors genetic diversity at a level typically found between species of the same genus.

**Suggested mechanism of genetic diversity generation.** Our observations are consistent with multiple chromosomes within cells frequently recombining and undergoing rearrangement, possibly through homologous and non-homologous recombination and transposition. The latter is plausibly responsible for the variability in gene synteny observed in genomic and metagenomic data of *Achromatium* cells from Lake Stechlin. Over 540 different transposases were identified in the metagenome, and up to 90 in single cells, several times more

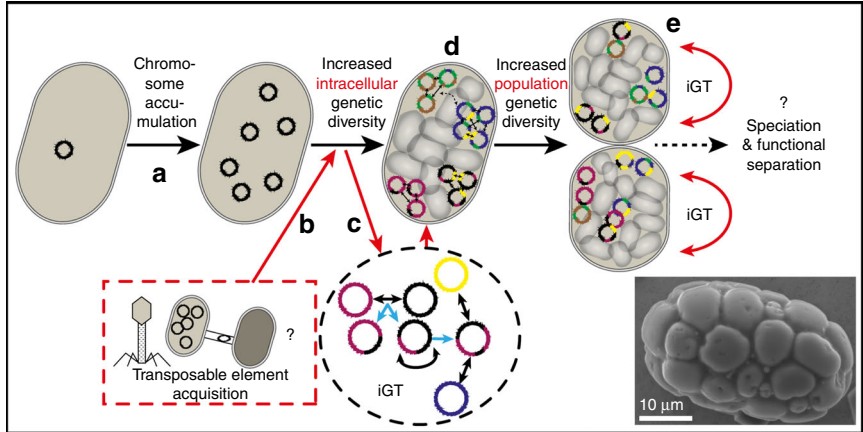

**Fig. 8** Proposed model for generation of large intracellular and subsequent population genetic diversity. Genome replication in an ancestral *Achromatium* cell **a** is followed by the acquisition of an unusually high number of transposases via undetermined mechanisms (e.g., phages, horizontal gene transfer); **b**. The genomes in the polyploid cell undergo a process of intracellular gene transfer (iGT) with limited transfer between cell parts **c** due to the multiple calcite bodies which can be seen in the insert. The new chimeric genomes further interact increasing the cellular diversity while maintaining conserved regions such as operons intact **c**. Overall this results in a cell with multiple different chromosomes; **d**. represented by the different colors of each chromosomes. Gene conversion does not occur on a cellular level but maintains genome stability at a local scale and thus fixing multiple version of the same gene in different cellular locations. A dividing cell recruits part of the genomic material generating two daughter cells different from each other as well as from the mother cell; **e**. Scale bar in SEM insert is 10 μm

than other *Achromatium* species (Table 1; Supplementary Data 7). This large intra-genus difference is not unprecedented. Sequencing of six *Crocosphaera* isolates from two different phenotypic groups found one isolate containing over 1200 transposases. Other isolates from the same phenotype had up to 165 transposases, while isolates from the other phenotype had up to 223[38]. In contrast to previously studied *Achromatium* genomes[12, 15], insertion sequences (IS)–often encoding the transposase genes[39] were found in high abundance in our single cell as well as metagenomic data of Lake Stechlin. The number of identified ISs in the similarly annotated genomes of the brackish water *Achromatium*, however, was far lower (Table 1). Increased numbers of IS elements have been associated with environmental stress[40, 41], possibly enhancing genetic diversity by mutagenesis.

Transposable elements can explain the intracellular gene shuffling and part of the observed increased divergence. However, an additional mechanism is needed to explain the large genetic diversity in Lake Stechlin (and saltwater) *Achromatium* cells. In fact, a main disadvantage of polyploid, asexual organisms, is that new mutations affecting a few chromosome copies within a cell are thought to have limited phenotypic consequences, and so mutations that would be strongly deleterious if fixed within a cell, can be tolerated for many generations[42]. However, these mutations can go to fixation many generations after their occurrence in a cell, leading to a sudden decrease in fitness (replication load). A way to escape this replication load can be strong gene conversion leading new mutations to either fixation or extinction within cells. This mechanism is not likely to occur in *Achromatium* at the whole cell level, because it would lead to very little within-cell diversity, as observed, e.g., in *Epulopiscium*[9] and *Haloferax volcanii*[6]. Nevertheless, as visible in Fig. 1 and Supplementary Fig. 1, the *Achromatium* chromosomes (DNA spots) occur in separate clusters. Thus, gene conversion as well as evolutionary selection might take place in separate genetic hotspots (containing clusters of chromosomes) that are minimally or not at all interacting with each other. This would lead to fixation of different neutral and positive mutations in each "compartment". Cellular rearrangement upon cell division would lead to the propagation of these mutations and the fixation of new ones. The Ka/Ks analysis (Fig. 4b) shows that

most of the proteins analyzed are under evolutionary pressure. This suggests that a gene-conversion-like process might occur, however, on a local (compartment) rather than a global (whole cell) scale.

Another mechanism for polyploid prokaryotes to escape extinction is between-cell recombination (or horizontal gene transfer (HGT)), that could reintroduce beneficial variants lost within cells[42]. HGT can occur via conjugation, viral infection, gene-transfer-agents and the uptake of naked DNA from the environment. We only found the genes required for the latter across the *Achromatium* single-cell genomes and metagenome (e.g., the *comEA* genes and the assembly genes for type IV pili). Thus, while *Achromatium* might have the ability to uptake genetic material from the surrounding environment, HGT between adjacent cells is most likely not the major mechanism leading to the large observed genetic variation.

We propose that following a process of genome replication and accumulation in an ancestral cell of the modern *Achromatium* (Fig. 8a), the acquisition of multiple transposases (Fig. 8b) led to a continuous, intracellular exchange of genomic sequences between individual genomes of the polyploid cell (i.e., intracellular gene transfer (iGT); Fig. 8c). This resulted in a mosaic-like genome (Fig. 8d) consisting of conserved regions (genes and operons, e.g., Fig. 8d, e), which are separated by less conserved spacers. A possible indication of a viral source for these transposases can be found in numerous of the KilA-N domain containing proteins[43]. The spatial compartmentalization of these genomes alongside possible uptake of DNA from the environment, allows for the formation of multiple stable versions of similar proteins. We further suggest that the genomic diversity of single cells leads to a highly heterogenous community as dividing cells, which likely split into two genetically different (yet functionally similar) entities (Fig. 8e). Comparing the metagenomic data with the DoriC database[44], we identified 23 potential origins of replication (Table. 1). Interestingly, this number is similar to the number of different copies of the *dnaN* (DNA polymerase III beta subunit; 22 copies) gene that is found adjacent to the OriC and the replication initiator *dnaA*, of which 28 copies have been identified (Supplementary Data 7). Among the single cells <6 potential OriC were identified per genome both in the freshwater and

saltwater cells. It has already been proposed that some bacteria harbor more than one origin of replication on a single chromosome[45] and this has also been successfully achieved by genome engineering[45]. Thus, it is highly plausible that different genomes (chromosomes) in a single polyploid cell (such as *Achromatium*) harbor different origins of replication. This may allow the cell to regulate the number of different chromosomes present at any given time and subsequently, if different chromosomes harbor different proteins or protein copies with different properties, the cells may be able to tune their expression by enhancing the available number of DNA templates.

**Evolutionary and ecological significance of iGT**. This proposed mechanism of genome evolution (iGT) has multiple implications for our understanding of microbial evolution and the role of polyploidy. First, as our comparative analysis shows, high intracellular genomic diversity is not a common feature among all members of the *Achromatium* genus (i.e., saltwater vs. freshwater). Nevertheless, a higher than expected diversity among replicate copies of proteins within a cell is observed across the lineage. Interestingly, this is not uniformly depicted by the rRNA marker gene. More genome sequencing from different freshwater and marine environments is needed to understand the full extent of this phenomenon.

It is suggested that one benefit of polyploidy is to allow for the generation of "experimental" versions of functioning proteins (neofunctionalization)[46, 47]. As long as one (or more) functional copy of a given protein is present, cells can putatively allow divergence to accumulate in alternative copies leading to new specialized functions. The presence of a high number (up to almost 100) of multiple different copies of the same gene in our metagenomic data set suggests that this process might be predominant in *Achromatium*. It remains to be determined which and how many of such gene copies are indeed functional. Interestingly, we also detected multiple copies of group II introns (Supplementary Data 7), a feature that has been shown in the polyploid *Thiomargarita*[48], hinting towards eukaryotic-like alternative splicing.

Thus, can *Achromatium* cells represent an intermediate evolutionary state between uni- and multicellular life? Multicellular bacteria differ from their eukaryotic counterparts. Often these are chained single cells (e.g., filamentous *Cyanobacteria* and the large sulfur bacterium *Beggiatoa*) each of which can survive alone and give rise to a new filament. In contrast, different cells in a multicellular eukaryotic organism have various functions, a phenomenon which is rare in the prokaryotic domains (exceptions are, for example, heterocysts of $N_2$-fixing filamentous cyanobacteria[49]). *Achromatium* cells appear to have multiple compartments[12]. Inside of those, chromosomes might be independently replicated, and may serve as the basis of functional compartmentalization and multicellularity. Functional compartmentalization has been described in several bacteria[50] and compartment-specific gene expression has been described in spore-forming *Bacillaceae*[51]. At present, no indication exists as to differential expression of genes in different parts of the *Achromatium* cell. Some of the FISH images (Fig. 2b, Supplementary Figs. 2 and 6) suggest that some expression patterns may exist, but this needs further evaluation in more targeted studies. Additionally, Markov and Kasnacheev[42] suggest that, similarly to Lake Stechlin *Achromatium*, the proto-Eukaryote cell may have been a rapidly mutating, chromosome diversifying, polyploid Bacteria/Archaea.

To conclude, we present evidence for increased intracellular genomic diversity in single cells of the freshwater, large sulfur bacterium *Achromatium* from Lake Stechlin. We propose

that intra and inter-cellular gene transfer (iGT and HGT), chromosomal rearrangement, and compartmentalized gene conversion are responsible for this phenomenon. This is supported by the high abundance of transposable elements and the presence of multiple, mostly conserved, versions of the same proteins. Elevated genetic diversity might provide *Achromatium* with exceptional potential for fast adaptation. The extent and functional significance of this phenomenon in permanently or temporarily polyploid bacteria remains to be determined.

## Methods

**Cell collection**. Samples were collected from Lake Stechlin in Germany (53.1520 °N, 13.0233 °E) on several occasions during 2015 and 2016. Surface sediment from the lake's shore at a water depth of ca. 1 m were allowed to settle for a few hours in a glass beaker, after which the surface layer was sieved through a 200 μm hole size mesh onto a glass plate. Single cells were picked under a binocular and transferred to a clean vial by using the different sedimentation of *Achromatium oxaliferum* cells and sand grains when applying a rotational movement. The cells were further transferred until no further coarse contaminants were visible. See details below for single-cell processing.

*Achromatium* cells are covered in a layer of extracellular polymeric substances to which epibiotic bacteria were attached. To remove this layer and thus minimize foreign DNA contamination, the concentrated cells were washed for 10 min in a 100 mM $NaHCO_3$ buffer.

Cells for DNA extraction were frozen at −20 °C until further treatment. Samples for fluorescent in situ hybridization were fixed for 14 h at 4 °C in 1% formaldehyde solution (final concentration).

**Fluorescence in situ hybridization and DNA staining**. DNA staining with SybrGreen I or PicoGreen was performed on non-fixed cells with a mixture of 5 μl of the stock solution in 200 μl Sigma Mowiol plus 5 μl freshly prepared 1 M ascorbic acid in 1× TAE buffer.

For fluorescence in situ hybridization, fixed *Achromatium* cells were washed with distilled water allowing the cells to precipitate prior to water removal. The clean, fixed cells were placed in polylysin covered microscope slides. Fluorescence in situ hybridization was done as previously described[52] using a 10% and 30% formamide hybridization buffer for the Achro664 and AchroCL probes respectively. Hybridization was carried out for 12–14 h at 46 °C. The probes Achro664I (5′-gcttggctagagtacgaaag-3′), Achro664II (5′-gccaagctagagtacgaaag-3′) and Achro664III (5′-gctgagctagagtacgggag-3′) were designed using the full-length 16S rRNA sequences obtained from the metagenome. The probes were labeled with 6-FAM, Cy-3, and Cy-5, respectively, and were generated by Biomers (Ulm, Germany). The probes AchroCLI (5′-ggatcgtcgccttggtaggcca-3′), AchroCLIII (5′- cgatcgttgccttggtgggctt-3′), and AchroCLII-IV (5′-cgatcgtcgccttggtaggctt-3′) were designed based on the V1–V4 regions of the 16S rRNA gene and start at position 263. The probes were labeled with Cy-3, Cy-5, and 6-FAM, respectively, and were generated by Biomers (Ulm, Germany) and were doubly labeled for enhanced signal.

Images were taken using an Axiovision microscope (Zeiss). An autofluorescence signal was observed in all channels although it required exposure times for image acquisition at least 10 times longer than for the FISH signal. Hybridization experiments using the EUB (I,II,III) probe mix and the non-EUB probe produced, positive and negative signals, respectively.

**Scanning electron microscopy**. The scanning electron microscopy image was taken using a Jeol JCM-6000 using 15 kV at high-vacuum at a magnification of X1500. The *Achromatium* cells were dried for 3 h 30 °C, placed in an exetainer for 15 h, and were sputtered with Au-Pd for 5 min.

**Metagenomics**. The fragile nature of *Achromatium* was used to extract DNA without additional chemical steps. Circa 10,000 cells were placed in 500 μl water and disrupted on a vortex machine for 30 s after which they were centrifuged in a table top centrifuge (Fresco 17, Thermo Fisher) for 5 min at 17,000×g. The DNA was precipitated with 2 vol. of 29:1 ethanol:sodium acetate (3 M) at −20 °C overnight followed by 30 min centrifugation at 17,000×g at 4 °C. The DNA pellet was washed with ice cold 70% ethanol, centrifuged at 17,000×g for 5 min, dried, and re-suspended in water.

Sequencing was carried out at Molecular Research Laboratories (Mr. DNA), Shallowater, Texas on an Illumina HiSeq resulting in 96,000,000 paired end reads (2 × 151 nucleotides). Metagenome sequencing steps included DNA fragmentation, ligation to sequencing adapters, and purification. Following the amplification and denaturation steps, libraries were pooled and sequenced. DNA (50 ng) from each sample was used to prepare the libraries using Nextera DNA Sample Preparation Kit (Illumina). Library insert size was determined by Experion Automated Electrophoresis Station (Bio-Rad). The insert size of the libraries ranged from 300 to 850 bp (average 500 bp). Pooled library (12 pM) was loaded to a 600 Cycles v3

Reagent cartridge (Illumina) and the sequencing was performed on HiSeq (Illumina).

Raw sequence data was quality trimmed using the Nesoni Clip tool (http://www.vicbioinformatics.com/software.nesoni.shtml). To remove the non-uniform coverage caused by the genome amplification step the samples were normalized to a coverage of 100-fold using the BBnorm tool (https://sourceforge.net/projects/bbmap/). The normalized sequences were assembled using SPAdes[53] version 3.7 including the implemented error correction tools. Note that due to SPAdes being specifically designed for assembly of single-cell data, no major differences in assembly were obtained when the normalized and raw data were used.

The assembled data were binned using Metawatt[24] version 3.5.2. Bins not associated to *Chromatiaceae* were not used for further analysis. A similar binning approach was applied to four published genomes of *Achromatium*[12, 15]. Metawatt was also used to extract the sequences of all identified single-copy marker genes as well as partial 16S rRNA sequences.

The combined bins associated with *Achromatium* were analyzed using the Prokka pipeline[54]. Selected bins or bin clusters (as underlined in Fig. 4) were also uploaded to the IMG-ER[31] system for annotation.

Percent completion of the single cells genomes as well as of the metagenomic bins alongside duplication level were analyzed using Metawatt[23]. These data were confirmed by analysis of the same data using CheckM[55] with the Chromatiaceae marker-genes set, which also provided the estimated genome size.

For the analysis of coding sequences, all copies of individual genes/proteins were extracted and aligned using MUSCLE[56]. Trees showing the diversity within each protein/gene copy were calculated using FastTree[57] version 2.1. Specific trees were calculated using RAxML[58]. Distance analysis and non-synonymous/synonymous ratio were calculated using the megacc command line version of MEGA[59] version 7. Distances between trees of proteins were calculated using the TreeDist program of the PHYLIP package.

16S rRNA sequence analysis from the raw metagenomics data were conducted using phyloFlash (https://github.com/HRGV/phyloFlash).

**Single-cell genomics.** *Achromatium* cells were collected as mentioned above in two separate occasions. In the first attempt, single cells were placed in a 2 ml tube containing 15 μl of MilliQ water. Five cells were selected for sequencing based on differences in morphology or cells size: a small, medium, and large sized cell as well as two dividing cells. Cells were sequenced at Molecular Research Laboratories (Mr. DNA), Shallowater, Texas, where DNA was extracted by centrifugation. Thanks to the fragile nature of the cells they break when centrifuged allowing the genomic DNA to spill out. Genomic DNA was amplified using the REPLI-g Single Cell kit (Qiagen).

Single-cell genomes were obtained as above; however, the reads were dominated (ca. 90%) by epibionts.

The DNA was then used for 16S rRNA amplicon sequencing on the Illumina MiSeq platform. The 16S rRNA gene V1-3 variable region PCR primers 28f/515r[60] with barcodes on the forward primer were used in a 28 cycle PCR (five cycles used on PCR products) using the HotStarTaq Plus Master Mix Kit (Qiagen, USA) under the following conditions: 94 °C for 3 min, followed by 28 cycles of 94 °C for 30 s, 53 °C for 40 s, and 72 °C for 1 min, after which a final elongation step at 72 °C for 5 min was performed. After amplification, PCR products were checked by agarose gel electrophoresis to determine the success of amplification and the relative intensity of bands. Multiple samples were pooled (e.g., 100 samples) in equal proportions based on their molecular weight and DNA concentrations. Pooled samples were purified using calibrated Ampure XP beads and the purified PCR product used to prepare the Illumina DNA library. Sequencing was performed following the manufacturer's guidelines.

In a second attempt to obtain clean genomes of single *Achromatium* cells, 50 NaHCO₃ treated single cells were collected into separate 2 ml tube prefilled with sterile PCR grade water. Each cell was then transferred 3 times into a new similar 2 ml tube. Last, each cell was transferred to a new empty tube in a total volume of ~1 μl water. Of these 25 cells were selected for whole-genome amplification using the Illustra Single Cell GenomiPhi DNA Amplification Kit (GE Healthcare Europe GmbH, Freiburg, Germany) resulting in 22 successful amplification which were sent for 16S rRNA gene sequencing at Molecular Research Laboratories (Mr. DNA), Shallowater, Texas using a PGM machine with primer set 515F-806R. The resulting sequences were analyzed using the DADA2 R[19] package. Six cells with >99% of their 16S rRNA gene sequences attributed to *Achromatium* were selected for single-cell genome sequencing. The sequencing was carried out as described above.

Single-cell genomes were assembled using SPADES[53] (V 3.9) and binned using Metawatt[24]. Genomic bins associated with *Chromatiaceae* were extracted, the raw reads were mapped to the contained contigs and reassembled. The final scaffolds were annotated using Prokka[54], RAST[61], and IMG-ER[31].

**Amplicon data analysis.** The 16S rRNA amplicon sequences were analyzed in full by three independent pipelines: the proprietary pipeline of the sequencing company based on QIIME[62], the SILVA NGS pipeline[63], and the DADA2 R package[19]. For the first, sequences were joined, depleted of barcodes, and sequences < 150 bp or with ambiguous base calls were removed. Sequences were denoised, OTUs generated, and chimeras removed. OTUs were defined by

clustering at 3% divergence (97% similarity). Final OTUs were taxonomically classified using BLASTn against a curated database derived from RDPII and NCBI (www.ncbi.nlm.nih.gov; http://rdp.cme.msu.edu). The SILVA pipeline was supplied with a FASTA file containing the assembled reads as provided by the sequencing company.

The analysis using the DADA2 R package[19] was conducted using the raw reads and the default parameters.

The *Achromatium* representative sequences together with all available 16S rRNA *Achromatium* sequences in the database were aligned using the online version of SINA[64]. Phylogenetic trees were calculated using RAxML (v 8.2.8)[58], the GTR model, and 100 bootstrap analyses.

**Data availability**. All sequence data are available at the European Nucleotide Archive (ENA) under project number PRJEB14545/ERP016191 (available through the ENA ftp site). Annotated data can be accessed on IMG genome IDs 2642422595-9, 2710724216-17, 2711768587-90 and on MG-RAST under id mgp11828.

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

## Acknowledgements

The positions of D.I. and M.B.-I. were financed through the DFG Aquameth (GR 1540/21-1) and BMBF BIBS projects. We thank Dieter Babenzien for the introduction to Achromatium ecology and advice on cell collection. Further, we acknowledge Lara Sabelhaus for collecting and cleaning cells, Katrin Attermeyer for graphical assistance, Reingard Rossberg for SEM images, Sina Schorn, Verena Salman-Carvalho, David Walsh, Mark Dopson, Wolfgang Hess and Alicia Muro-Pastor for fruitful discussions and for critically commenting on this manuscript.

## Author contributions

D.I.: concept, data generation, data analysis, and wrote the paper. M.B.-I.: data generation, data analysis, and wrote the paper. N.D.M.: data analysis and wrote the paper. H.C.: concept, data generation, and wrote the paper. H.-P.G.: concept, data generation, and wrote the paper.

## Additional information

**Competing interests:** The authors declare no competing financial interests.

