## [Peer Review File · Nature Communications]

Reviewers' comments:

Reviewer #1 (Remarks to the Author):

Ionescu et al. ask truly fascinating questions about the potential polyploidy and intracellular gene shuffling in freshwater sediment bacteria *Achromatium*. These phenomena, if verified, would have significant implications to our understanding of bacterial evolution and ecology. However, I have major reservations about the conclusions of the study, due to poorly described and likely flawed methods. Most of the data presented in the manuscript comes from metagenomic assemblies and subsequent contig binning. This type of data is not suited to discriminate genetic variation among versus within individual cells. The only single cell sequence data that is discussed in any length is 16S rRNA gene heterogeneity, which can be easily explained by alternative scenarios, e.g. multiple and non-identical 16S genes being encoded on a single genome copy - a common and well described feature in many bacteria. Furthermore, it is disappointing that the authors provide no details on the critical techniques that were employed to separate individual cells and to verify that they indeed were single cells (line 607 in Materials and Methods). Even the microscopy evidence for multiple genome copies per cell (Fig. 1) is tenuous: instead of using a dsDNA-specific stain (e.g. picoGreen) the authors applied SYBRGreen I, which also stains RNA, thus making it difficult to discriminate chromosomes from ribosomes and other nucleic acids-containing entities. These methodological limitations put into question the key conclusions of the manuscript.

Reviewer #2 (Remarks to the Author):

The manuscript by Ionescu et al. describes single cell genomics of *Achromatium* and gives evidence that the genomes in the polyploid cells are not identical, but in contrast, exhibit a diversity normally typical of whole communities. The possibility that highly polyploid giant bacteria might contain a mixture of genomes has been discussed previously. However, attempts to find evidence for genome variability in single cells or filaments in *Epulopiscium* and *Candidatus Maritrix* sp. indicated that all genomes were extremely similar or identical. Therefore, the report by Ionescu et al. gives the first evidence for intracellular genomic variability. Clearly, this is of outstanding importance and will be of high interest for a widespread readership!!! I congratulate the authors to such an important manuscript.

However, I have a few points that should be addressed.

1) Line 31. I suggest to remove the last part of the sentence ("..., aiding its survival..."). It has not been shown that Lake Stechlin is an over-average dynamic environment that needs high evolution rate for survival. In addition, it is not clear whether intracellular genome diversity increases fitness, it may well decrease fitness and lead to the eventual extinction of this *Achromatium* lineage.

2) L. 51. Salman-Carvalho et al. (2016, *Front Microbiol.* 7: 1173) should be added as a second example.

3) L. 75f. The authors argue that the varying fluorescence intensities are indicative of "varying amount of DNA, i.e. different number of chromosomes". That might well be true, however, other explanations seem possible. 1) the saturation with the stain might be different for different genomes (I could not find the staining procedure in Material and Methods, probably fixed cells were used). 2) If the genome variability is really caused by the high number of transposons, as proposed by the authors, the result should be genomes of different sizes. This is not discussed at all in the manuscript and should be added at appropriate places. It might be interesting to analyze the distribution of integrated fluorescence intensities of the intracellular spots to see whether they form a continuum or have discrete values. In addition, from the volume of the *Achromatium* cell it could be calculated how many genome copies would be expected if it had the genome density (per volume) as *Epulopiscium* or *Bacillus*.

4) L. 29, L. 98, Figure 2B. The authors argue that the staining is indicative of "spatially-differential" gene expression. I find the expression a bit misleading because it seems to indicate

that the same gene has a different expression level at different sites within one cell. It has indeed been suggested that for giant bacteria diffusion is limited and thus the extra- and intracellular concentrations of substrates and metabolites differ locally, resulting in spatially-differential gene expression of identical genomes. In contrast, in the current manuscript it is argued that the genomes are not identical, therefore, (putative) identical gene expression levels of different rRNA genes lead to spatially-differential ribosome distributions, which are shown in Fig. 2B. These two different meanings of "spatially-differential" should be clarified. To visualize the spatial-differential localization of ribosomes even better, I would propose to make Figure 2B a separate Figure and include overlays of A-C, A-B, and B-C. It should also be discussed why the left half of the cell does not seem to contain A, B, or C ribosomes.

5) L. 157f. *recA* is shown in Fig. 3E (not D), and *rplK* is shown in Fig. 3D (not E). Is the conserved gene neighborhood only seen for "all ribosomal proteins", or also for other conserved operons, e.g. the *atp*- or *trp* operon?

6) L. 223ff. In my opinion Muller's ratchet is not "such a mechanism", in stark contrast, it predicts that cells with intracellular genomic diversity should not be able to exist. A mechanism that can explain genomic diversity is the absence of gene conversion in this strain of *Achromatium*. Markov and Kaznacheev (2016) should be cited here, who have modeled the instability of polyploid species without a high level of gene conversion. While "neutral and beneficial mutations may accumulate alongside deleterious ones", the number of deleterious mutations will be higher than the number of beneficial mutations. Therefore, for me the mechanism of escaping Muller's ratchet for *Achromatium* is not clear.

7) L. 245-250. This is extremely speculative without any experimental indication and should be marked as such. The 23 origins might as well be overpredictions or they might all be identically present on all chromosomes.

8) L. 281-297. Again, this paragraph is very speculative. Fig. 1 does not show multiple invaginations of the cytoplasmic membrane and functional compartmentalization on the way to multicellularity. Instead, the cell seems to divide into two cells of equal size. Not "quite similar" but rather different, the *in silico* model of Markov and Kasnacheev proposes the invention of mitosis as an escape of Muller's ratchet for a unicellular polyploid prokaryote with diversified chromosomes in the absence of sex.

9) Fig. 1. B and D seem to be identical.

10) Fig. 3D. It seems odd that in only 1-2 of the 9 genomic sites there are genes at the left and the right side. I would propose to remove 10-20% from the left and the right and enlarge the middle 60-70% to better visualize the synteny around *rplK*.

11) Fig. 3E. I would suggest to show *recA* in the identical direction in all sites (like *rplK* in Fig. 3D). Why is *recA* in the third site from top so much smaller? Please add a scale to Fig. 3D and E. Are the many light yellow genes really meant to be orthologous?

12) Fig. 3E shows an *in silico* reconstruction of single cell sequencing data. Single cell PCR with primers for *recA* and different neighboring genes would have been a very convincing experimental validation showing that the proposed variable genomic compositions really exists in one cell (and as all sites have multiple copies per cell, this should be experimentally feasible).

Reviewer #3 (Remarks to the Author):

Ionescu and colleagues investigated the intracellular phylogenetic diversity of the large sulfur bacterium *Achromatium* using metagenomic and single-cell genomic sequencing, along with fluorescent *in situ* microscopy. The authors demonstrate *Achromatium* is a polyploid bacterium, harboring multiple chromosome copies per cell as evidenced through microscopy imaging. 16S rRNA analyses from 5 single cells that underwent genome amplification suggest nearly 2,000 clusters at 97% identity, far greater genetic diversity than what would be expected for a single coherent bacterial species. Further, the metagenome data was binned to examine putative assembled genomes and resulted in presumably discordant genomes with multiple copies of known single-copy conserved genes and an unusually high number of encoded transposases. With this

data, the authors speculate about the mechanisms of genetic diversity within *Achromatium* and hypothesize about the evolutionary significance of polyploidy.

Overall, the study is conceptually interesting considering the potential role polyploidy might play in genome evolution for this large sulfur bacterium and challenges conventional wisdom regarding genetic divergence within microbial species. That said, I found many aspects of the experimental design flawed. Further, the authors in many cases over-interpreted the data and propose a mechanism for genetic diversification that is not entirely supported by the data. The authors should revisit their data to validate that the "diversity" observed is not due to methodological artifact (MDA and sequence error, see below). It would also be highly advised to include additional data for a single-cell *Achromatium* genome that is not highly contaminated with epibiont DNA to provide a more defined genomic reference. The sequence data presented in the current manuscript is of insufficient quality to truly infer the level of genetic diversity associated with polyploidy.

Single cell data. The authors indicate five single cells were selected for MDA amplification, and then due to the high contamination by epibiont DNA (~90%) the amplified single cells underwent 16S amplicon sequence. This would introduce significant amplification biases in the resultant 16S data. MDA alone introduces significant biases (see Zhang 2015 Nature Comm as a recent example). From just five single cells, the authors report a total 1989 OTUs clustered at 97% identity, the generally accepted threshold for species demarcation. Were the clusters evaluated at lower thresholds that might collapse this seemingly high diversity? The authors indicate that alignment of the clustered OTUs showed variations concentrated in the hypervariable regions and interpret this as a result of evolutionary processes and not sequencing error. However, the amplicons span three hypervariable regions and thus this interpretation is weak. Could the authors provide estimates for sequencing error and demonstrate empirically that the sequence errors do not account for the variation?

Metagenome data. The metagenome bins are questionable given the poor assembly of the metagenome (longest contigs <56 kbp). I'm surprised to see any bins that could be accurately identified from such short assembled contigs. Tetranucleotide frequencies for contigs shorter than 2kb are not considered an accurate estimate for a genomic fragment to represent a whole-genome level kmer frequency, and thus the results from the binning are questionable. While the sequencing coverage should have been sufficient coverage as the authors suggest, it is curious why the data did not assemble better. Could there be additional pre-processing to improve the assembly? Perhaps additional sequencing using long read technology (PacBio) would enable improved genomic recovery. Regardless, identifying accurate bins using such fragmented assembly would be impossible. It is therefore not surprising that the authors found duplicate (and in some cases multiple) copies of conserved "single copy" genes – these bins are likely highly contaminated. The authors do not provide any information about estimated heterogeneity across the bins, which I suspect is high. There are tools available (checkM, AMPHORA, ect) to calculate these estimates (along with % genome completeness). It is not clear how genome completeness and contamination was assessed since no information about the genome bins is provided. A supplementary table including the number of bins, size of bins (# contigs, total #bp), whether they contain rRNA genes, and # of conserved single-copy marker genes. It's a pity the metagenome is not of better quality since the single cell data did not result in recovery of the *Achromatium* genome.

16S phylogeny. It is not valid to generate phylogenetic trees with fragments of the 16S rRNA gene from the amplicon data. A more appropriate way to generate a robust phylogenetic tree is to build the alignment using the full-length 16S rRNA sequences and add in the shorter amplicon sequences using a parsimony method. The authors additionally need to provide specific details on how the tree was constructed with the number of alignment positions and exact evolutionary model.

Microscopy. The FISH imaging is very compelling and is the best line of evidence that

Achromatium is polyploidy. It is interesting the apparent localization of the probes specific to the three phylotypes identified. While I don't advocate for additional experiments since they would be quite laborious, it would be exciting to quantify this property across multiple cells to see if this property is conserved or just a transient state of the cell. This would be the first example (to my knowledge) of a bacterium capable of compartmentalizing genetically distinct copies of the chromosome.

Throughout the manuscript, the authors appear to exaggerate the current state of the field. For example: Pg. 2, line 44: The statement that the role of polyploidy has "been almost completely ignored in Bacteria and Archaea" is an overstatement given the work on *Deinococcus radiodurans*, *Thermus thermophilus*, *Epulopiscium* spp. and many haloarchaea, to name a few. In this instance, I would suggest the authors shift the emphasis to reflect that the role of polyploidy in the bacteria and archaea has been underexplored. The authors should carefully edit the manuscript to avoid sensationalism.

Response to reviewers

Reviewers' comments:

Reviewer #1 (Remarks to the Author):

Ionescu et al. ask truly fascinating questions about the potential polyploidy and intracellular gene shuffling in freshwater sediment bacteria *Achromatium*. These phenomena, if verified, would have significant implications to our understanding of bacterial evolution and ecology. However, I have major reservations about the conclusions of the study, due to poorly described and likely flawed methods. Most of the data presented in the manuscript comes from metagenomic assemblies and subsequent contig binning. This type of data is not suited to discriminate genetic variation among versus within individual cells. The only single cell sequence data that is discussed in any length is 16S rRNA gene heterogeneity, which can be easily explained by alternative scenarios, e.g. multiple and non-identical 16S genes being encoded on a single genome copy - a common and well described feature in many bacteria. Furthermore, it is disappointing that the authors provide no details on the critical techniques that were employed to separate individual cells and to verify that they indeed were single cells (line 607 in Materials and Methods). Even the microscopy evidence for multiple genome copies per cell (Fig. 1) is tenuous: instead of using a dsDNA-specific stain (e.g. picoGreen) the authors applied SYBRGreen I, which also stains RNA, thus making it difficult to discriminate chromosomes from ribosomes and other nucleic acids-containing entities. These methodological limitations put into question the key conclusions of the manuscript.

We have made the following changes to the manuscript to address the comments from the reviewer:

- 1) We have repeated the staining of the DNA using picoGreen as suggested by the reviewer. We have obtained identical results confirming the multiple DNA spots (genomes/chromosomes) in *Achromatium*.
- 2) We have sequenced the genomes of 6 single *Achromatium* cells and have integrated this information into the manuscript alongside the metagenomic data.
- 3) We have improved the clarity of the methods section.

Reviewer #2 (Remarks to the Author):

The manuscript by Ionescu et al. describes single cell genomics of *Achromatium* and gives evidence that the genomes in the polyploid cells are not identical, but in contrast, exhibit a diversity normally typical of whole communities. The possibility that highly polyploid giant bacteria might contain a mixture of genomes has been discussed previously. However, attempts to find evidence for genome variability in single cells or filaments in *Epulopiscium* and *Candidatus Marithrix* sp. indicated that all genomes were extremely similar or identical. Therefore, the report by Ionescu et al. gives the first evidence for intracellular genomic variability. Clearly, this is of outstanding importance and will be of high interest for

a widespread readership!!! I congratulate the authors to such an important manuscript.

However, I have a few points that should be addressed.

1) Line 31. I suggest to remove the last part of the sentence (“..., aiding its survival...”). It has not been shown that Lake Stechlin is an over-average dynamic environment that needs high evolution rate for survival. In addition, it is not clear whether intracellular genome diversity increases fitness, it may well decrease fitness and lead to the eventual extinction of this Achromatium lineage.

By dynamic environment we referred to the ever perturbed coastal environments which are much more affected by storms for example than deeper sediments, but we agree that this was not clear. To prevent misunderstandings we have now removed this statement from the abstract.

2) L. 51. Salman-Carvalho et al. (2016, Front Microbiol. 7: 1173) should be added as a second example.

Thank you for pointing out this missing reference. It was now added to the manuscript.

3) L. 75f. The authors argue that the varying fluorescence intensities are indicative of “varying amount of DNA, i.e. different number of chromosomes”. That might well be true, however, other explanations seem possible. 1) the saturation with the stain might be different for different genomes. 2) If the genome variability is really caused by the high number of transposons, as proposed by the authors, the result should be genomes of different sizes. This is not discussed at all in the manuscript and should be added at appropriate places. It might be interesting to analyze the distribution of integrated fluorescence intensities of the intracellular spots to see whether they form a continuum or have discrete values. In addition, from the volume of the Achromatium cell it could be calculated how many genome copies would be expected if it had the genome density (per volume) as Epulopiscium or Bacillus.

We indeed consider the possibility of size difference between the different chromosomes as highly plausible. However, it seems that this was not clearly stated in the original version. We have better addressed this issue in the corrected version.

The quantification of fluorescence intensity is unfortunately not reliable. Due to the thickness of the cells the generated stacked image results in lower fluorescence from deeper spots than from higher ones regardless of their DNA content.

Achromatium also poses a challenge in the estimation of genome density. While the volume of the cell can be easily calculated, it does not reflect the volume of the cytoplasm. There, at least 70% (and probably more) is occupied by calcium bodies and the presence of additional vacuoles is suspected.

4) L. 29, L. 98, Figure 2B. The authors argue that the staining is indicative of “spatially-differential” gene expression. I find the expression a bit misleading because it seems to indicate that the same gene has a different expression level at different sites within one cell. It has indeed been suggested that for giant bacteria diffusion is limited and thus the extra- and intracellular concentrations of substrates and metabolites differ locally, resulting in spatially-differential gene expression of identical genomes. In contrast, in the current manuscript it is argued that the genomes are not identical, therefore, (putative) identical gene expression levels of different rRNA genes lead to spatially-differential ribosome

distributions, which are shown in Fig. 2B. These two different meanings of “spatially-differential” should be clarified.

Line 98 has been modified to state that in some cases different alleles of the same genes may be spatially-differentially expressed in different locations in the cell.

To visualize the spatial-differential localization of ribosomes even better, I would propose to make Figure 2B a separate Figure and include overlays of A-C, A-B, and B-C. It should also be discussed why the left half of the cell does not seem to contain A, B, or C ribosomes.

This figure has been replaced with a new FISH analysis used improved FISH probes. We have followed the reviewer’s suggestion and have shown all combinations of overlaid images.

5) L. 157f. *recA* is shown in Fig. 3E (not D), and *rplK* is shown in Fig. 3D (not E). Is the conserved gene neighborhood only seen for “all ribosomal proteins”, or also for other conserved operons, e.g. the *atp*- or *trp* operon?

Following the reviewer’s suggestion, we have added information about these additional two operons. The addition of the new single cells provided some interesting insights into the topic. Based on scaffolds that are long enough to contain the entire operon, the ATP operon is not consistently conserved. Interestingly, in at least 3 cases in which most genes are present the operon is interrupted by an RNase and RNase-inhibitor couple (the same in all cases).

Using the TRP operon model from the salt water *Achromatium* as a reference we expect the beta subunit of the tryptophan synthase to be followed by the alpha one. Such operons are present in the freshwater *Achromatium* as well. In most cases the alpha subunit is followed by Acetyl-CoA carboxylase carboxyltransferase. The latter is almost consistently following the alpha chain of the tryptophan synthase. Thus, we hypothesize that when the scaffold is too short on the 5’ end the beta chain is expected to be present as well. However, we have found several cases where either the alpha or beta chains were present independently with the beta chain being followed by transposase domains or Endonucleases

6) L. 223ff. In my opinion Muller’s ratchet is not “such a mechanism”, in stark contrast, it predicts that cells with intracellular genomic diversity should not be able to exist. A mechanism that can explain genomic diversity is the absence of gene conversion in this strain of *Achromatium*. Markov and Kaznacheev (2016) should be cited here, who have modeled the instability of polyploid species without a high level of gene conversion. While “neutral and beneficial mutations may accumulate alongside deleterious ones”, the number of deleterious mutations will be higher than the number of beneficial mutations. Therefore, for me the mechanism of escaping Muller’s ratchet for *Achromatium* is not clear.

We have modified this section and have in fact removed the discussion regarding Muller’s ratchet. We speculate that compartmentalization caused by the multiple calcite bodies leads to genomic pockets (which at least organizationally can be seen in Fig.1) with limited recombination between them, but possible recombination within them. We have redrawn our model (Fig. 8) to represent this idea. This mechanism allows for the formation of several stable and functional versions of the same protein.

Another mechanism to escape Muller's ratchet could be between-cell recombination, however we find no evidence of the machinery to allow the transfer of DNA between cells (i.e. conjugation).

7) L. 245-250. This is extremely speculative without any experimental indication and should be marked as such. The 23 origins might as well be overpredictions or they might all be identically present on all chromosomes.

We have emphasized that these are potential and predicted OriC.

8) L. 281-297. Again, this paragraph is very speculative. Fig. 1 does not show multiple invaginations of the cytoplasmic membrane and functional compartmentalization on the way to multicellularity. Instead, the cell seems to divide into two cells of equal size. Not "quite similar" but rather different, the in silico model of Markov and Kasnacheev proposes the invention of mitosis as an escape of Muller's ratchet for a unicellular polyploid prokaryote with diversified chromosomes in the absence of sex.

We have now modified this section of the discussion.

9) Fig. 1. B and D seem to be identical.

These two panels are indeed similar. Panel D has an additional layer showing the DNA spots counted by the software.

10) Fig. 3D. It seems odd that in only 1-2 of the 9 genomic sites there are genes at the left and the right side. I would propose to remove 10-20% from the left and the right and enlarge the middle 60-70% to better visualize the synteny around rplK.

This figure has now been completely remade.

11) Fig. 3E. I would suggest to show recA in the identical direction in all sites (like rplK in Fig. 3D). Why is recA in the third site from top so much smaller? Please add a scale to Fig. 3D and E. Are the many light yellow genes really meant to be orthologous?

This figure has been completely remade. RecA has been removed from the figure. Following the suggestion to look into other operons we now show the ribosomal operon as well as the two suggested operons ATP synthase and Tryptophan synthase. This has been done since synteny should have a higher functional significance in operons rather than solitary genes. As the new figure shows not all operons are conserved though there is evidence that "standard" copies occur in the cells.

12) Fig. 3E shows an in silico reconstruction of single cell sequencing data. Single cell pcr with primers for recA and different neighboring genes would have been a very convincing experimental validation showing that the proposed variable genomic compositions really exists in one cell (and as all sites have multiple copies per cell, this should be experimentally feasible).

The high sequence variability within these cells, as can be seen from the ANI analysis and the phylogenetic trees of the single copy marker genes, makes it hard to design reliable primers. Nevertheless, we believe that the new data coming from 6 single cells further showcases the lack of gene synteny as well as synteny-altering mechanisms such as transposable elements or other genes that seem to have been repeatedly inserted in some loci.

Reviewer #3 (Remarks to the Author):

Ionescu and colleagues investigated the intracellular phylogenetic diversity of the large sulfur bacterium *Achromatium* using metagenomic and single-cell genomic sequencing, along with fluorescent in situ microscopy. The authors demonstrate *Achromatium* is a polyploid bacterium, harboring multiple chromosome copies per cell as evidenced through microscopy imaging. 16S rRNA analyses from 5 single cells that underwent genome amplification suggest nearly 2,000 clusters at 97% identity, far greater genetic diversity than what would be expected for a single coherent bacterial species. Further, the metagenome data was binned to examine putative assembled genomes and resulted in presumably discordant genomes with multiple copies of known single-copy conserved genes and an unusually high number of encoded transposases. With this data, the authors speculate about the mechanisms of genetic diversity within *Achromatium* and hypothesize about the evolutionary significance of polyploidy.

Overall, the study is conceptually interesting considering the potential role polyploidy might play in genome evolution for this large sulfur bacterium and challenges conventional wisdom regarding genetic divergence within microbial species. That said, I found many aspects of the experimental design flawed. Further, the authors in many cases over-interpreted the data and propose a mechanism for genetic diversification that is not entirely supported by the data. The authors should revisit their data to validate that the “diversity” observed is not due to methodological artifact (MDA and sequence error, see below). It would also be highly advised to include additional data for a single-cell *Achromatium* genome that is not highly contaminated with epibiont DNA to provide a more defined genomic reference. The sequence data presented in the current manuscript is of insufficient quality to truly infer the level of genetic diversity associated with polyploidy.

We have now added whole-genome sequencing data of 6 new single cells. These cells were selected following a preliminary sequencing of 16S rRNA gene from 25 cells, based on having over 99% of the amplicons affiliated with *Achromatium*.

Single cell data. The authors indicate five single cells were selected for MDA amplification, and then due to the high contamination by epibiont DNA (~90%) the amplified single cells underwent 16S amplicon sequence. This would introduce significant amplification biases in the resultant 16S data. MDA alone introduces significant biases (see Zhang 2015 Nature Comm as a recent example). From just five single cells, the authors report a total 1989 OTUs clustered at 97% identity, the generally accepted threshold for species demarcation. Were the clusters evaluated at lower thresholds that might collapse this seemingly high diversity? The authors indicate that alignment of the clustered OTUs showed variations concentrated in the hypervariable regions and interpret this as a result of evolutionary processes and not sequencing error. However, the amplicons span three hypervariable regions and thus this

interpretation is weak. Could the authors provide estimates for sequencing error and demonstrate empirically that the sequence errors do not account for the variation?

We are now in the possession of two single-cell 16S rRNA amplicon data sets. The first set, presented in the original manuscript, was analyzed de-novo using the DADA2 software which deduces true variants from sequencing errors. This reduced the data to 20 OTUs rather than the previous 1189. The second set sequenced using ion-torrent was analyzed in a similar manner and produced 177 OTUs instead of ~7000 that would result from normal distance based clustering. In our new phylogenetic analysis presented in Fig. 2 and a supplementary figure we have used the sequences representing these lower estimates of diversity

Metagenome data. The metagenome bins are questionable given the poor assembly of the metagenome (longest contigs <56 kbp). I'm surprised to see any bins that could be accurately identified from such short assembled contigs. Tetranucleotide frequencies for contigs shorter than 2kb are not considered an accurate estimate for a genomic fragment to represent a whole-genome level kmer frequency, and thus the results from the binning are questionable. While the sequencing coverage should have been sufficient coverage as the authors suggest, it is curious why the data did not assemble better. Could there be additional pre-processing to improve the assembly? Perhaps additional sequencing using long read technology (PacBio) would enable improved genomic recovery. Regardless, identifying accurate bins using such fragmented assembly would be impossible. It is therefore not surprising that the authors found duplicate (and in some cases multiple) copies of conserved "single copy" genes

– these bins are likely highly contaminated. The authors do not provide any information about estimated heterogeneity across the bins, which I suspect is high. There are tools available (checkM, AMPHORA, ect) to calculate these estimates (along with % genome completeness). It is not clear how genome completeness and contamination was assessed since no information about the genome bins is provided. A supplementary table including the number of bins, size of bins (# contigs, total #bp), whether they contain rRNA genes, and # of conserved single-copy marker genes. It's a pity the metagenome is not of better quality since the single cell data did not result in recovery of the *Achromatium* genome.

We are indeed aware that the bins do not represent genomes of individual cells, as also confirmed by phylogenetic inconsistencies between marker genes from multiple bins, and we now make this clearer from the text. We now show that genomes amplified from single cells also lead to such bins, confirming that the bins do not represent clear biological subdivisions. In addition to removing non *Achromatium* contaminants, we use the binning to emphasize the lack of consensus in the genomic and metagenomic data, thus supporting our hypothesis of multiple and variable genomes within individual cells and within the population.

A supplementary table with bin information has been added (Supplementary Data S1). This table includes bin/genome completion data, duplication level and estimated genome size. The data was generated by Metawatt and CheckM.

With respect to assembly quality of the metagenome – we have tried numerous assemblers, believing at first that this is an assembly issue. The SPADES assembler proved to give the best results (longest scaffolds). The assembly did not improve by using the newer versions of SPADES (3.6-3.10) or using

SPADESmeta. A similar result was obtained from the single cell data where long contigs could not be obtained. The addition of the 6 new single cells to the metagenomic data did not change the assembly statistics as neither pooling of the 6 cells into one data set.

We think that the formation of only short contigs is a result of having “small” conserved islands connected by variable regions. Due to the extreme polyploidy (possibly higher than coverage) and within-host variation, we suspect that it is not possible to build a consensus for non-conserved regions.

16S phylogeny. It is not valid to generate phylogenetic trees with fragments of the 16S rRNA gene from the amplicon data. A more appropriate way to generate a robust phylogenetic tree is to build the alignment using the full-length 16S rRNA sequences and add in the shorter amplicon sequences using a parsimony method. The authors additionally need to provide specific details on how the tree was constructed with the number of alignment positions and exact evolutionary model.

Using RAxML as implemented in ARB we built a tree using the 17 full length sequences available in the database and have added the new data using parsimony and a position variability filter of 10%. Since partial sequences outnumber available references, some branches cannot be inferred with great confidence. We have added the parsimony tree as a supplementary figure (Fig S3). We believe it conveys a similar message as our de-novo created tree.

Microscopy. The FISH imaging is very compelling and is the best line of evidence that Achromatium is polyploidy. It is interesting the apparent localization of the probes specific to the three phlotypes identified. While I don't advocate for additional experiments since they would be quite laborious, it would be exciting to quantify this property across multiple cells to see if this property is conserved or just a transient state of the cell. This would be the first example (to my knowledge) of a bacterium capable of compartmentalizing genetically distinct copies of the chromosome.

We have repeated the experiment multiple times both with the FISH probes from the original manuscripts as well as with newly design probes that are presented in this version. The strong separation observed in the original image occurs yet is not the most common. As can be seen in the new figure the probe signal do not fully overlap but they are not always strongly separated into distant areas of the cell.

Throughout the manuscript, the authors appear to exaggerate the current state of the field. For example: Pg. 2, line 44: The statement that the role of polyploidy has “been almost completely ignored in Bacteria and Archaea” is an overstatement given the work on *Deinococcus radiodurans*, *Thermus thermophilus*, *Epulopiscium* spp. and many haloarchaea, to name a few. In this instance, I would suggest the authors shift the emphasis to reflect that the role of polyploidy in the bacteria and archaea has been underexplored. The authors should carefully edit the manuscript to avoid sensationalism.

We have rephrased these sentences to better acknowledge the work done in the field.

REVIEWERS' COMMENTS:

Reviewer #1 (Remarks to the Author):

The addition of whole genome analyses of individual cells is the most significant improvement of the manuscript since the prior submission. This provides compelling evidence for extensive, intracellular genomic heterogeneity, e.g. non-identical copies of what is considered "conserved single-copy genes" inside individual cells. However, I have concerns over the interpretation of these new data:

1. On lines 128-141, the authors claim that the absence of single cell genome contigs >82 kb constitutes a proof of polyploidy. In fact, this level of assembly fragmentation is typical for single cell genomics and is primarily caused by the uneven multiple displacement amplification. Therefore short contig size cannot be used as a proof of polyploidy.

2. On lines 165-167, in contrast to authors' claims, single cell genomics data does not and cannot prove or disprove the described metagenomic results. Metagenome bins are by their definition not suited for intracellular genomic heterogeneity studies. The discussion of metagenomic bins should be revised accordingly throughout the paper.

3. Lines 286-288: The claim that horizontal gene transfer is not significant in *Achromatium* is unsubstantiated and should be either supported by empirical evidence or removed.

Other comments:

- Fig. 1: Although nucleic acid imaging was re-done with a DNA-specific stain, this more robust analysis is still not presented in the main body of the manuscript (Fig. 1) and no method description was provided. Imaging of intracellular DNA distribution is by far the most important evidence of polyploidy in *Achromatium*, so making sure this evidence is compelling is really important.

- Fig. 3: A legend explaining each pane is needed.

- Fig. 4: I have absolutely no idea what this figure shows, because its legend does not match with the content.

- Fig. 6: The most compelling type of finding here would be diversity of genomic neighborhoods of the same gene in the same cell. Unfortunately, I don't see any such examples here. One exception may be tryptophan synthase in cell 14, but data quality is questionable, due to frequent misassemblies close to contig ends. Are there better examples to be presented here?

- Line 216: "Dayhoff distance" needs an explanation.

- Line 254: Term "phenotype" is misused.

- Lines 246-312: this discussion is overly speculative. It needs to be supported by the actual data or removed.

Reviewer #2 (Remarks to the Author):

The authors have reacted adequately to my remarks as well as - in my opinion - the remarks of the other reviewers. Specifically, they have sequenced six additional single cells and integrated the results into the manuscript, they have used additional operons for the synteny analysis, and remade several Figures. In my opinion the manuscript can be published in the present form.

only:

- correct transposase to transposases in Table 1
- Define A - H in the Legend to Figure 3
- The legends A and B to Figure 4 seem to be swapped

Reviewer #3 (Remarks to the Author):

The authors should be commended for their thorough and rigorous work to acquire new data and validate their results in the revised manuscript. The present work is significantly improved and

well-organized. A few minor comments below, but otherwise the authors have addressed all previous concerns. The work is quite novel and represents an important contribution to the field.

The concept of “genomic pockets” is intriguing and could explain the evolutionary trajectory of genomic diversity within single *Achromatium* cells. Suggest including this concept in the abstract to highlight the novelty.

In the abstract, the sentence “...show that individual cells of the world biggest known freshwater bacterium...” should be revised to “...show that individual cells of the world’s largest known freshwater bacterium...”

Pg 5. Line 199 – The authors indicate “data not shown, available on IMG.” The authors should provide the appropriate IMG identifier or accession for these datasets within the IMG database. Also, a more updated reference to IMG should be included (<https://doi.org/10.1093/nar/gkw929>).

Reviewer #1 (Remarks to the Author):

The addition of whole genome analyses of individual cells is the most significant improvement of the manuscript since the prior submission. This provides compelling evidence for extensive, intracellular genomic heterogeneity, e.g. non-identical copies of what is considered “conserved single-copy genes” inside individual cells. However, I have concerns over the interpretation of these new data:

1. On lines 128-141, the authors claim that the absence of single cell genome contigs >82 kb constitutes a proof of polyploidy. In fact, this level of assembly fragmentation is typical for single cell genomics and is primarily caused by the uneven multiple displacement amplification. Therefore short contig size cannot be used as a proof of polyploidy.

We do not claim that the short contigs are a proof of polyploidy but rather of increased genomic diversity, which allows for a good assembly of the relatively conserved regions. We further claim that less conserved regions binding the obtained contigs vary too much to be covered and accordingly assembled by the available data from our sequence analyses and from the databases.

2. On lines 165-167, in contrast to authors’ claims, single cell genomics data does not and cannot prove or disprove the described metagenomic results. Metagenome bins are by their definition not suited for intracellular genomic heterogeneity studies. The discussion of metagenomic bins should be revised accordingly throughout the paper.

Our discussion on metagenomic bins has two purposes. First, the splitting of single-cell data into multiple bins, which do not represent different *Achromatium* species, supports the fact that also the metagenomic binning does not result from multiple species but rather from a similar intracellular genomic complexity across multiple cells. Second, the splitting of single cell data into multiple bins each containing different versions of what are supposed to be “single copy” marker genes, in our opinion, supports the presence of multiple versions of the *Achromatium* genome. We do not claim that these bins represent stable groupings of the different genomic versions, nor do we believe such exist. We hope that this is obvious for the reader when reading our text.

3. Lines 286-288: The claim that horizontal gene transfer is not significant in *Achromatium* is unsubstantiated and should be either supported by empirical evidence or removed.

Since horizontal gene transfer (HGT) between cells (via conjugation) is a known mechanism for increasing genetic diversity we felt compelled to address it. Based on the absence of relevant genes for the conjugation mechanisms in our metagenomic and single-cell sequence data, we hypothesize that this is not a likely reason for the large observed intracellular diversity. We have modified the text to clearly relate to HGT via conjugation and to state that this is probably and most likely not the case for our *Achromatium* cells.

Other comments:

- Fig. 1: Although nucleic acid imaging was re-done with a DNA-specific stain, this more robust analysis is still not presented in the main body of the manuscript (Fig. 1) and no method description was provided. Imaging of intracellular DNA distribution is by far the most important evidence of polyploidy in *Achromatium*, so making sure this evidence is compelling is really important.

The results of the PicoGreen and SybrGreen staining don't differ. Hence, we chose to leave the original image which depicts a dividing cell.

The method used for both PicoGreen and SybrGreen staining is given in the methods section: "DNA staining with SybrGreen I or PicoGreen was performed on non-fixed cells with a mixture of 5 μ l of the stock solution in 200 μ l Mowiol plus 5 μ l freshly prepared 1 M ascorbic acid in 1x TAE buffer."

- Fig. 3: A legend explaining each pane is needed.

The legend was amended accordingly.

- Fig. 4: I have absolutely no idea what this figure shows, because its legend does not match with the content.

This figure as described in the caption and further evaluated in the text shows phylogenetic trees of the bins obtained from the metagenomic data as well as the single cell genome data. Trees were calculated with the binning software Metawatt. In our opinion, these trees are part of the supporting evidence for the large intracellular genetic diversity of *Achromatium*, which is also reflected by the metagenomic data. Thus, similar to the single cell data, the metagenomic data does not result from multiple *Achromatium* species but rather from one species with a rather large genetic variability on the single cell level.

- Fig. 6: The most compelling type of finding here would be diversity of genomic neighborhoods of the same gene in the same cell. Unfortunately, I don't see any such examples here. One exception may be tryptophan synthase in cell 14, but data quality is questionable, due to frequent misassemblies close to contig ends. Are there better examples to be presented here?

We did not evaluate the neighborhoods of all genes but rather of several operons where gene synteny can be expected to be conserved. In these cases, there were cells exhibiting more than one copy of these genes, but often these were on smaller contigs from which no information on neighboring genes were available. Accordingly, those genes are not presented. The data is nevertheless openly available on IMG under the genome ids given in the text. It is obvious from the figure that gene synteny in the operons is not well conserved between the single cells and, interestingly, is often interrupted by similar genetic components.

- Line 216: "Dayhoff distance" needs an explanation.

A reference was added for the Dayhoff protein evolutionary model: "Dayhoff, M. O., Schwartz, R. N. & Orcutt, B. C. A model of evolutionary change in proteins. *Atlas Protein Seq. Struct.* 5, 345 (1978)"

- Line 254: Term “phenotype” is misused.

The term phenotype in this case is directly taken from the title of the cited paper: “Bench, S. R. et al. Whole genome comparison of six *Crocospaera watsonii* strains with differing phenotypes. J. Phycol. 49, 786–801 (2013)” and thus we believe that we have used it in the right manner.

- Lines 246-312: this discussion is overly speculative. It needs to be supported by the actual data or removed.

The reviewer suggests removing a substantial portion of our discussion. However, we don't aim to present our discussion as absolute facts but rather as our own interpretation of the data. We provide several lines of support for our interpretation, which allow the reader to follow our logic. Since all the data is available, any reader including the reviewer are free to reach their own conclusions and interpretations of our dataset. However, we believe that our discussion provides guidance for further experiments to be planned and carried out to prove or disprove the here presented hypotheses.

Reviewer #2 (Remarks to the Author):

The authors have reacted adequately to my remarks as well as - in my opinion - the remarks of the other reviewers. Specifically, they have sequenced six additional single cells and integrated the results into the manuscript, they have used additional operons for the synteny analysis, and remade several Figures. In my opinion the manuscript can be published in the present form.

Thank you for your positive statement.

only:

- correct trasnposase to transposases in Table 1

Corrected

- Define A - H in the Legend to Figure 3

The panels were defined.

- The legends A and B to Figure 4 seem to be swapped

The legend was corrected accordingly.

Reviewer #3 (Remarks to the Author):

The authors should be commended for their thorough and rigorous work to acquire new data and validate their results in the revised manuscript. The present work is significantly improved and well-organized. A few minor comments below, but otherwise the authors have addressed all previous concerns. The work is quite novel and represents an important contribution to the field.

Thank you for your positive statement.

The concept of “genomic pockets” is intriguing and could explain the evolutionary trajectory of genomic diversity within single *Achromatium* cells. Suggest including this concept in the abstract to highlight the novelty.

We added the concept to the abstract within the length limitations of the journal.

In the abstract, the sentence “...show that individual cells of the world biggest known freshwater bacterium...” should be revised to “...show that individual cells of the world’s largest known freshwater bacterium...”

Corrected

Pg 5. Line 199 – The authors indicate “data not shown, available on IMG.” The authors should provide the appropriate IMG identifier or accession for these datasets within the IMG database. Also, a more updated reference to IMG should be included (<https://doi.org/10.1093/nar/gkw929>).

The IMG ids were added to the text. We have also updated the reference of IMG to the latest paper as suggested by the reviewer.